# Giant spin ensembles in waveguide magnonics

Zi-Qi Wang[1], Yi-Pu Wang [1]✉, Jiguang Yao[1], Rui-Chang Shen[1], Wei-Jiang Wu[1], Jie Qian[1], Jie Li[1], Shi-Yao Zhu[1] & J. Q. You [1]✉

The dipole approximation is usually employed to describe light-matter interactions under ordinary conditions. With the development of artificial atomic systems, 'giant atom' physics is possible, where the scale of atoms is comparable to or even greater than the wavelength of the light they interact with, and the dipole approximation is no longer valid. It reveals interesting physics impossible in small atoms and may offer useful applications. Here, we experimentally demonstrate the giant spin ensemble (GSE), where a ferromagnetic spin ensemble interacts twice with the meandering waveguide, and the coupling strength between them can be continuously tuned from finite (coupled) to zero (decoupled) by varying the frequency. In the nested configuration, we investigate the collective behavior of two GSEs and find extraordinary phenomena that cannot be observed in conventional systems. Our experiment offers a new platform for 'giant atom' physics.

Since the wavelength of the light is substantially larger than the size of the atoms, atoms and quantum emitters are naturally considered as point-like objects in typical light-matter interaction systems, such as a hundred-nanometer-wavelength optical field interacting with angstrom-scale atoms[1,2]. Surprisingly, a new paradigm of 'giant atom' develops in the artificial atomic system, where superconducting qubits are exploited as giant atoms[3–9]. The self-interference among the different parts of the giant atom gives rise to numerous exotic phenomena that are unachievable in small atoms, such as the frequency-dependent relaxation rate[3,8], non-exponential decay of the giant atom[4,6], and decoherence-free interaction between giant atoms[5,8]. Similar effects have also been reported with superconducting qubits (small atoms) placed close to the end of a transmission line[10,11]. Despite being a mesoscopic quantum system, the superconducting qubit[12,13] is still smaller than the wavelength of the resonant microwave in a general configuration. Thus, the early 'giant atom' research was carried out in the system of superconducting qubits coupled to short-wavelength surface acoustic waves[4,6,14–17], where the sound wavelength is small enough to go beyond the dipole approximation[3]. Nevertheless, the core of the 'giant atom' physics is that the emitter cannot be regarded as a point, so it can also be simulated as an atom interacting with the optical field at multiple points. In such a way, superconducting qubits can be realized as giant atoms by coupling it to a meandering waveguide at separated points[8,18].

The interaction between a single giant atom and light gradually becomes experimentally accessible, but more insightful physics is incorporated in the interaction between multiple giant atoms[5,9,19,20]. Meanwhile, a variety of intriguing theoretical schemes have recently been proposed[19–24] by exploiting the exotic interference effect in the 'giant atom' physics, which are anticipated to be experimentally validated. However, the experimental studies of solid-state devices in the giant-atom regime have been only limited to superconducting-qubit systems. With the increasing number of superconducting qubits, the experimental difficulty is greatly raised. Until now, only braided two artificial giant atoms have been demonstrated in the superconducting-qubit system at cryogenic temperature[8]. In this sense, a flexible and easy-to-operate experimental system is beneficial to fully evaluate and investigate the 'giant atom' physics.

The study of 'giant atom' should not be limited to atomic systems, but can be extended to other spin systems and harmonic oscillator systems. In the microwave region, the ferromagnetic spin ensemble is an easily tunable system with the benefits of flexible and wide-range adjustability of the magnon mode frequency (a few hundred megahertz to several tens of gigahertz), low dissipation at room

[1]Interdisciplinary Center of Quantum Information, State Key Laboratory of Modern Optical Instrumentation, and Zhejiang Province Key Laboratory of Quantum Technology and Device, School of Physics, Zhejiang University, 310027 Hangzhou, China. ✉e-mail: yipuwang@zju.edu.cn; jqyou@zju.edu.cn

temperature[25–29], and extendibility by constructing the magnon-based hybrid system[30–34]. In order to further reveal the unique physics beyond the dipole approximation, we construct the giant spin ensemble by coupling the ferromagnetic spin ensemble [yttrium iron garnet (YIG) sphere] to a meandering waveguide at two coupling points. The distance between the coupling points is set to be larger than the resonance wavelength (several centimeters) of the magnon mode, which is much larger than the size (millimeter) of a YIG sphere.

In this work, we focus on the demonstration of the GSE setup and the collective effects in the nested GSEs configuration. We show that not only the interaction strength but also the interaction mechanism between nested GSEs are frequency-dependent, which can be tuned from purely coherent coupling to dissipative coupling. The outer GSE acts as a cavity in the designed structure. In the dissipative coupling case, two GSEs cooperatively dissipate to the waveguide without direct interaction. In the coherent coupling case, although the outer GSE is completely decoupled from the waveguide, an evident coherent behavior between the two GSEs mediated by the waveguide is unexpectedly observed. The GSE and waveguide magnonics are expected to serve as an ideal platform for investigating 'giant atom' physics. What's more, the nested configuration provides a new way to observe coherent behavior in an open environment, which shows a great potential for quantum information manipulation in waveguide QED.

## Results

We construct the GSE in the manner illustrated in Fig. 1a, where the spin ensemble interacts with the microstrip waveguide at two well-separated locations. The spin collective excitation mode (magnon mode) interacts with traveling photons at these two positions, giving rise to the propagating phase of the photons in this non-local coupling configuration. The phase is proportional to the distance between two coupling points, i.e., the effective size $L$ of the giant spin ensemble, and can be expressed as

$$\varphi = \omega_{\mathrm{m}} \frac{L}{v},\tag{1}$$

where $\omega_{\mathrm{m}}$ denotes the frequency of the magnon mode in the ensemble and $v$ is the microwave speed propagating through the waveguide. In the experiment, the waveguide is meandering in shape, and the mitered corners are designed for smooth microwave transmission over a wide frequency range, as shown in Fig. 1b. To illustrate the 'giant atom' physics, the distance between the two coupling points of the inner (outer) sphere is designed to be 8.3 (16.6) cm, all greater than the wavelength (about 2–4 cm) of the microwave they interact with. The spin ensembles we use are 1 mm diameter single-crystal YIG spheres, which are commercially available and manufactured by Ferrisphere Inc. (http://www.ferrisphere.com). Two YIG spheres $S_{\mathrm{o}}$ (green) and $S_{\mathrm{i}}$ (blue) are nested on the waveguide as shown in Fig. 1b, c.

The experiment is carried out at room temperature. A bias magnetic field is applied to tune the magnon mode frequency. The magnon mode is the collective spin excitation mode, and it can be treated as a harmonic oscillator mode which can have sufficiently large excitations. The thermal excitations at room temperature will not affect the interaction between the magnon mode and microwave photon mode in the waveguide. When the spin ensemble is uniformly magnetized and its magnetization is saturated, the magnon mode frequency is linearly proportional to the bias field [Fig. 1d], $\omega_{\mathrm{m}} = \gamma(H_{\mathrm{e}} + H_{\mathrm{A}})$, where $\gamma/2\pi = 28$ GHz/T is the gyromagnetic ratio, $H_{\mathrm{e}}$ and $H_{\mathrm{A}}$ are the bias magnetic field and anisotropy field, respectively. The bias field is supplied by an electric magnet and the power of the probe signal is −10 dBm. When only the $S_{\mathrm{i}}$ is placed, the experimentally measured reflection spectra versus bias field are shown in Fig. 1e. Two magnon

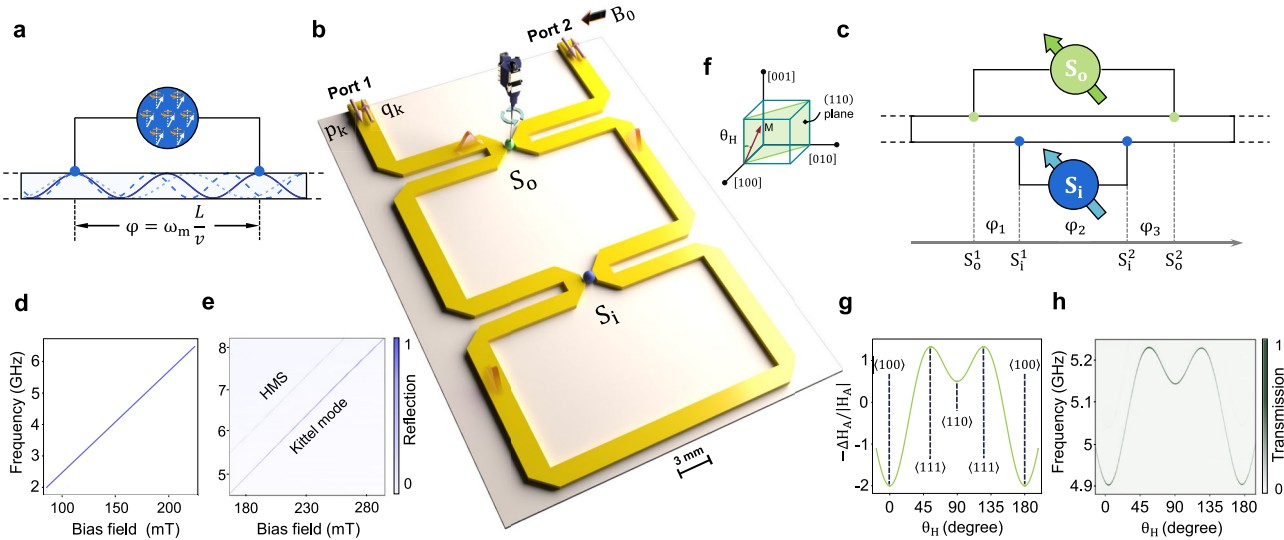

**Fig. 1 | Implementation of single giant spin ensemble and nested two giant spin ensembles. a** Schematic of a single GSE, where a small YIG sphere couples with the waveguide at two well-separated locations. The phase $\varphi$ induced by the propagation of the microwave photons between the two coupling points can be adjusted by tuning the resonance frequency $\omega_{\mathrm{m}}$ of the magnon mode. **b** Schematic of the experimental device of the nested two GSEs, where the inner YIG sphere (blue) is fixed on the waveguide, and the outer YIG sphere (green) is glued on a cantilever which is rigidly connected to a rotating motor. These YIG spheres are magnetized by a bias field $B_{\mathrm{O}}$. The GSE is realized by coupling the YIG sphere to the meandering waveguide (yellow) and sending a probe signal $\hat{p}_{k}(\hat{q}_{k})$ (red arrow) into the waveguide, where the microwave signal (pink) can interact with the spin ensemble more than once, yielding an effectively giant size of the spin ensemble. **c** The topology of

the nested two GSEs. The blue (green) dots denote the coupling points between the inner (outer) YIG sphere and the waveguide. For our symmetric design, the phase $\varphi_1$ equals $\varphi_3$. As we define $L_{\mathrm{o(i)}} = S_{\mathrm{o(i)}}^2 - S_{\mathrm{o(i)}}^1$, the propagating phase between the two coupling points of the outer sphere $S_{\mathrm{o}}$ (inner sphere $S_{\mathrm{i}}$) is $\varphi_{\mathrm{o(i)}} = \omega_{\mathrm{o(i)}} L_{\mathrm{o(i)}}/v$. **d** Theoretical curve of the magnon mode frequency versus the bias magnetic field. **e** The measured transmission mapping, where a Kittel mode and a higher-order magnetostatic mode appear. **f** Illustration of the crystal axes and lattice plane of the YIG. $\theta_H$ is the angle between the [001] axis and the bias field. **g** The magnetocrystalline anistropy field as a function of $\theta_H$. The curve is calculated by only considering the first-order anisotropic energy when rotating the YIG sphere in the (110) plane[25]. **h** The transmission mapping measured versus $\theta_H$.

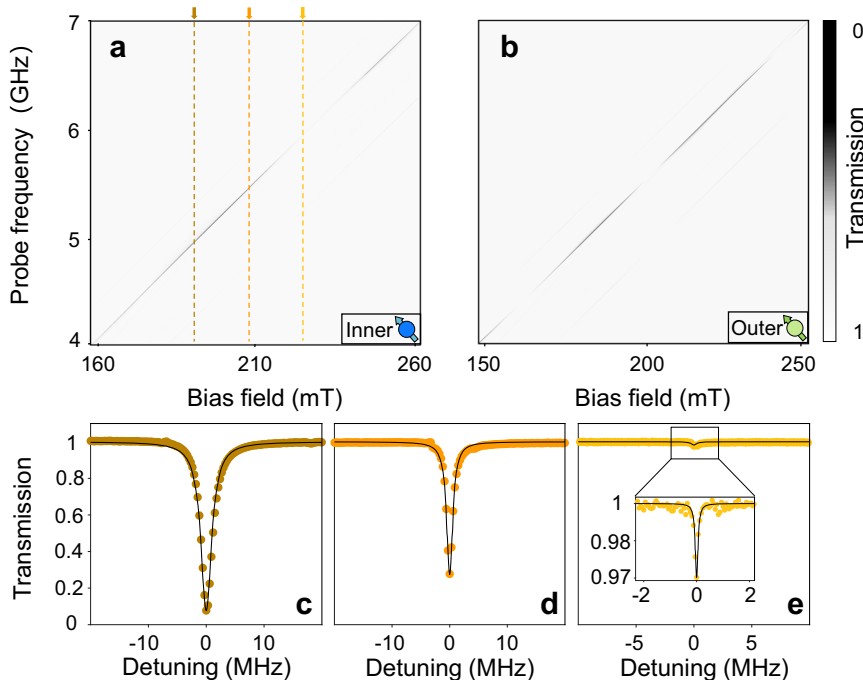

**Fig. 2 | Frequency-dependent radiative decay rate of a single giant spin ensemble. a, b** The transmission mapping plots of the inner (**a**) and outer (**b**) YIG spheres individually tuned by sweeping the bias field in the electric magnet. **c**–**e** Three typical spectra extracted from **a**. The dots are experimental data and the black solid curves are theoretical fittings.

modes are observed: one is the spin uniform precession mode, also known as the Kittel mode, and the other is a high-order magnetostatic mode (HMS). Our subsequent study only focuses on the Kittel mode.

We surround the entire sample with a large uniform bias field and fix the direction and intensity of the field. To clearly demonstrate the collective behavior of two GSEs via spectroscopy, the frequency of the magnon mode in one of the ensembles should be tuned individually. This is accomplished by gluing the outer sphere $S_o$ to the end of a cantilever attached to a rotating motor, as shown in Fig. 1b. We rotate the sample in its (110) plane during the experiment. The angle between the [001] crystal axis and bias magnetic field is defined as $\theta_H$, as depicted in Fig. 1f. The magnetocrystalline anisotropy field is determined by the angle, as shown in Fig. 1g[25]. By rotating the sphere $S_o$, we can individually tune the resonance frequency of the magnon mode in $S_o$. Here, we choose the spherical sample to circumvent the influence of the shape-related demagnetization field. The experimentally measured resonance frequency versus $\theta_H$ is depicted in Fig. 1h, which agrees well with the theory. We can see that the frequency of the Kittel mode in $S_o$ has a tuning range of about 330 MHz.

We begin with the experiment on a single GSE to examine the interference modulated effect in the inner and outer ensembles separately. We place the YIG sphere at the center of two mitered corners to ensure that the magnon mode is coupled to the waveguide with nearly equal strengths at both coupling points. In this case, the interaction Hamiltonian of a single magnon mode in one of the YIG spheres coupled with the meandering waveguide can be written as (see Supplementary Materials)

$$H_I/\hbar = g\left[\hat{a}_m^\dagger \hat{p}_k\left(e^{i\varphi}+1\right) + \hat{a}_m^\dagger \hat{q}_k\left(e^{-i\varphi}+1\right) + \text{h.c.}\right], \quad (2)$$

where $\hat{a}_m^\dagger$ is the magnon creation operator, $\hat{p}_k$ ($\hat{q}_k$) is the annihilation operator of the photon mode traveling from port 1 (2) to port 2 (1), and $g$ is the coupling strength between the spin ensemble and the waveguide at the coupling point, which can be expressed in terms of $\kappa$ (radiative damping rate at the coupling point) as $g = \sqrt{\kappa/2\pi}$ in the Markovian approximation.

It is clear from equation (2) that when $\varphi$ equals odd multiples of $\pi$, the spin ensemble decouples from the waveguide, because $e^{i\varphi} = e^{-i\varphi} = -1$. When fixing the effective size of the sample, we can see from equation (1) that the propagating phase $\varphi$ can be continuously adjusted by scanning $\omega_m$. In such a manner, the magnon mode will then periodically decouple from the waveguide.

First, we simply place the inner sphere $S_i$ on the waveguide and sweep the bias current of the electric magnet. The transmission mapping is shown in Fig. 2a, which consists of transmission spectra at varying bias field. It can be found that a transmission valley with periodic depth changes along the diagonal of the mapping plot, corresponding to the Kittel mode's resonance position in the inner spin ensemble. We can determine the damping rate of the magnon mode using spectroscopic fitting. Three typical transmission spectra at different magnetic fields are presented in Fig. 2c–e. We can clearly distinguish the difference in their linewidths. The broadest spectrum in Fig. 2c is due to the constructive interference between emissions from two coupling points. It is straightforward to deduce that the distance between two coupling points is equal to integer multiples of the wavelength in this case, i.e., the phase $\varphi$ equals even multiples of $\pi$. In Fig. 2d, we can find that the linewidth begins to decrease. The situation becomes obvious in Fig. 2e, where the signal of the magnon mode is obscured, indicating that the spin ensemble is almost decoupled from the waveguide. The radiations at two coupling points interfere destructively, and the phase $\varphi$ equals odd multiples of $\pi$. It is worth noting that the interference period should be inversely proportional to the effective size of the spin ensemble, i.e., $1/L$. Due to the fact that $L_o$ is larger than $L_i$, the outer GSE will periodically couple and decouple with the waveguide more quickly. As illustrated in Fig. 2b, we indeed observe a shorter interference period.

These transmission spectra can be well fitted by the following equation (see Supplementary Materials):

$$S_{21}(\omega) = 1 + \frac{\kappa_G}{i(\omega - \omega_m - \kappa\sin\varphi) - \kappa_G - \beta}, \quad (3)$$

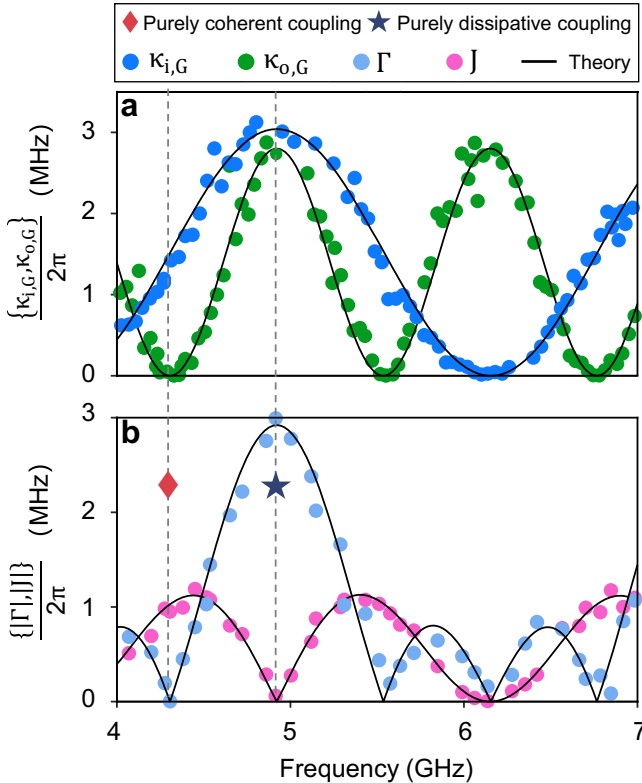

**Fig. 3 | Decay rates and coupling strengths of the nested two giant spin ensembles at different frequencies. a** The dots are the decay rates of the inner and outer GSEs measured at different frequencies. The solid curves are theoretical results obtained using Eq. (4). **b** The dots are the experimental data of the coherent interaction strength (pink) and dissipative interaction strength (light blue). The solid curves are theoretical results obtained using Eqs. (6) and (7). Two specific cases are labeled by pentagram and rhombus, which indicate that the coupling between the nested two spin ensembles are purely dissipative and purely coherent, respectively.

where the radiative decay rate of the GSE is

$$\kappa_G = 2\kappa\,(1 + \cos\varphi), \tag{4}$$

$\omega$ is the frequency of the probe field and $\beta$ is the nonradiative damping rate of the spin ensemble. Here, $\kappa_{i(o)}$ is defined as the radiative damping rate of the inner (outer) GSE at the point coupled to the waveguide, whereas $\kappa_{i(o),G}$ represents the radiative damping rate of the inner (outer) GSE acting as a giant atom, where the subscript G denotes 'giant'. By fitting the transmission spectra, we get $v = 3.26 \times 10^7$ m/s. The parameters of the outer GSE are $\kappa_o/2\pi = 0.70$ MHz, $\beta_o/2\pi = 1.39$ MHz, and $L_o = 16.56$ cm, while the parameters of the inner GSE are $\kappa_i/2\pi = 0.76$ MHz, $\beta_i/2\pi = 1.58$ MHz, and $L_i = 8.28$ cm. The radiative damping rates of the GSEs are plotted in Fig. 3a, where the periodic decoupling is clearly evident.

After demonstrating the characteristics of the individual GSE, we are more interested in the collective behavior of GSEs. Here, we specifically implement the nested configuration depicted in Fig. 1b, c, which has never been realized in experiment[9]. The propagating phase between the left coupling points of the outer sphere $S_o^l$ and the inner sphere $S_i^l$ is $\varphi_1$. Similarly, we define $\varphi_2$ and $\varphi_3$ as shown in Fig. 1c. It should be noted that $\varphi_1$ equals $\varphi_3$ in our device. There is no requirement that the two phases be equal in the actual situation. In the nested case, the effective interaction Hamiltonian of the two GSEs can be

written as (see Supplementary Materials)

$$H_N/\hbar = (J - i\Gamma)\left(\hat{a}_i^\dagger \hat{a}_o + \hat{a}_o^\dagger \hat{a}_i\right), \tag{5}$$

where $\hat{a}_{i(o)}^\dagger$ is the creation operator of the magnon mode in the inner (outer) YIG sphere and the dissipative coupling strength is

$$\Gamma = 2\sqrt{\kappa_i \kappa_o}\left(\cos\varphi_1 + \cos(\varphi_1 + \varphi_2)\right), \tag{6}$$

while the coherent interaction strength is

$$J = \sqrt{\kappa_i \kappa_o}\left(\sin\varphi_1 + \sin(\varphi_1 + \varphi_2)\right). \tag{7}$$

According to equations (5)–(7), the coupling strength is complex and can be adjusted via the propagating phases. We obtain a set of periodically varying coherent and dissipative coupling strengths by simultaneously tuning the frequencies of the magnon modes in the two YIG spheres, as shown in Fig. 3b. Among them, the interactions at 4.35 GHz and 4.96 GHz are particularly interesting, as indicated by the red rhombus and blue pentagram, since the coupling strengths are purely coherent and dissipative, respectively.

In the case of purely dissipative coupling, the propagating phase between the two coupling points of the inner (outer) sphere is an integer multiple of $2\pi$ [Fig. 4a], implying that the two GSEs dissipate maximally to the waveguide, as outlined in Fig. 3a. Additionally, the two coupling points of the outer GSE are equivalent to the two reflecting mirrors of a cavity, as demonstrated recently in a superconducting-qubit system[35]. As depicted in Fig. 4b, the two coupling points of the inner GSE are located at the equivalent cavity mode nodes, inferring that no coherent coupling occurs between the inner and outer GSEs, i.e., $J \approx 0$. Instead, they collectively dissipate to the waveguide, yielding a dissipative coupling between them. To observe the coupling behavior clearly, we fix the magnon mode frequency of the inner spin ensemble at 4.96 GHz using a global bias magnetic field and gradually rotate the outer spin ensemble to tune its frequency from 4.95 to 4.97 GHz. The measured transmission mapping is displayed in Fig. 4c, where the energy level attraction-like characteristic reflects dissipative coupling[36,37]. The result is consistent with our theoretical calculation, as depicted in Fig. 4d. As the consequence of dissipative coupling, the signature of the bright (superradiant) state is exhibited by the broadened linewidth, as shown in the inset graph of Fig. 4c.

It becomes even more fascinating when both the magnon modes in the two GSEs are tuned to 4.35 GHz. As shown in Fig. 4e, the propagating phase between the two coupling points of the outer GSE is odd multiples of $\pi$. The outer GSE no longer dissipates to the waveguide due to the destructive interference between the coupling points. The interference effect between the two coupling points of the inner sphere is in between complete constructive and destructive interferences, resulting in a finite dissipation, as shown in Fig. 3a. One might think that since the outer sphere is decoupled from the waveguide, the waveguide cannot mediate interaction between the two GSEs. However, the two coupling points of the inner sphere are no longer located at the nodes of the equivalent cavity formed by the outer GSE. The schematic model is represented in Fig. 4f. In the experiment, we observe the transmission mapping of energy level repulsion, as shown in Fig. 4g, which indicates the coherent coupling between the two GSEs (see theoretical result in Fig. 4h). The coupling strength $J/2\pi = 1.0$ MHz is obtained by fitting the hybridized polariton of the system at resonance. This counter-intuitive appearance cannot be reached with two small spin ensembles in the two-port waveguide system, when one of the spin ensembles is decoupled from the waveguide[38]. The coupling between the two GSEs is, in fact, in the magnetically induced transparency regime ($\kappa_{i,G} > J > \kappa_{o,G}$). Although this interaction is mediated by the open environment of the waveguide, our theoretical prediction

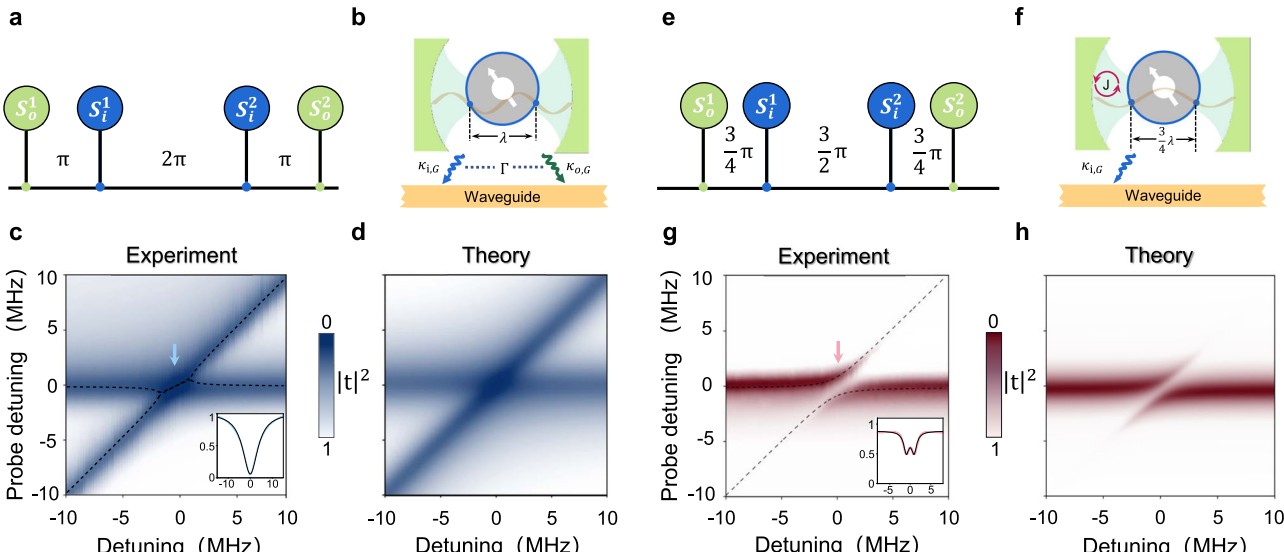

**Fig. 4 | The equivalent model of the nested two giant spin ensembles, and the spectroscopy studies. a, e** Sketch of the propagating phase between the two coupling points of the inner (outer) sphere in the cases of purely dissipative coupling (**a**) and purely coherent coupling (**e**), respectively. **b, f** Schematic of the analogous coupled system. The coupling points of the outer GSE work as two mirrors to form a cavity. The inner YIG sphere (blue) and the cavity collectively radiate into the waveguide (**b**), leading to the dissipative coupling between the inner and outer GSEs. The outer YIG sphere decouples from the waveguide (**f**), and coherent coupling between the inner and outer GSEs is obtained. **c, g** Transmission mappings measured by fixing the global bias field and rotating the outer YIG sphere. Energy level attraction-like spectra observed in the case of purely dissipative coupling (**c**) and the energy level repulsion observed in the case of purely coherent coupling (**g**). The black dashed curves correspond to the theoretically obtained real parts of the eigenvalues of the coupled system. The inset graphs depict the spectra at the resonant positions, which show the features of the broadened linewidth and the magnetically induced transparency, respectively.
**d, h** Theoretical calculation of the transmission mappings corresponding to **a** and **e**. The parameters used in the calculation are obtained in Fig. 2.

shows that it may enter the strong coupling regime, as long as $\kappa_o$ is large enough, that is $\sqrt{\kappa_o \kappa_i} > \max\{\kappa_{i,G} + \beta_i, \beta_o\}$. It should be noted that coherent coupling mediated by traveling-wave photons is feasible in a normal waveguide system, but strong coherent coupling is challenging since the coupling is mediated by a dissipative process. In this work, we see that the meandering waveguide together with the 'giant atom' physics may provide a promising method to realize strong coherent coupling between spin ensembles mediated by the traveling-wave photons. Similar approach has been realized in superconducting-qubit systems[8,35].

## Discussion

To summarize, we have experimentally realized the GSE and achieved periodic coupling and decoupling between the GSE and the waveguide by adjusting the frequency of the magnon mode in the spin ensemble. More importantly, we have also realized the nested structure of two GSEs. We quantitatively analyze the propagating phase between coupling points and the resulting periodic coherent and dissipative couplings. It agrees well with our theoretical model.

The self-interference effect, which is the defining characteristic of 'giant atom' physics, can be clearly shown in our system. In comparison with the 'giant atom' demonstrations in superconducting-qubit systems[6,8,18], the ferromagnetic spin system can exhibit a large frequency tuning range, simple device fabrication, and the ability to be constructed in a variety of topologies. It is an outstanding new platform for revealing and exploring novel phenomena related to the light-matter interaction beyond the dipole approximation.

From previous studies and ours, we can expect the future development of 'giant atom' physics in the following three aspects. First, it is expected to construct more novel microwave and optical devices base on the 'giant atom' physics. In the 'giant atom' system, the self-interference effect can be utilized and manipulated by tuning the phase difference between the coupling points, which gives rise to the periodic coupling and decoupling of the resonators or emitters. The phase difference can be controlled by the resonance frequency of the resonators

or emitters coupled to the meandering waveguide. When decoupling occurs, the transmission spectrum reveals maximum transmission, whereas the spectrum reveals very low transmission and considerably high reflection when the coupling is large. Therefore, we obtain a bandpass filter that is frequency-dependent and tunable. With the self-interference effect of 'giant atom' physics, it is promising to design more microwave and optical devices with a variety of functions.

The second aspect is related to the building of 'giant atom' networks. Beyond the dipole approximation, the self-interference effect can result in the realization of periodic coupling and decoupling, and the coupling between different giant atoms can also be regulated and become non-local. Using these features, we may construct a meandering waveguide-mediated network of giant atoms to realize a variety of information processing and storage schemes.

The third one is about demonstrating quantum effects in the 'giant atom' system. The phenomena addressed in this work only exploit the interference effect in a light-matter interaction beyond the dipole approximation, which has correspondences in both classical and quantum physics. For pure quantum effects, such as the quantum entanglement, vacuum squeezed state, etc., particularly the non-local effect of the quantum entanglement, one can expect to perform some interesting demonstrations in the context of 'giant atom' physics in the future.

In addition, the higher-order magnetostatic modes that are not discussed in the present work may be of interest in future study. We can harness these higher-order modes to increase the control degrees of freedom of the system and explore exotic phenomena in the multi-mode 'giant atom' system. Moreover, we can miniaturize the sample design by adding more bends along the waveguide or using a substrate with larger relative dielectric constant to fabricate the waveguide. Finally, we also look forward to the exploration of singular coupling behaviors, by combining topological[39] and non-Hermitian physics[40] with the 'giant atom' physics, which are expected to advance our ability of controlling the interaction between light and matter to another new level.

## Methods

### Device design

As illustrated in Supplementary Fig. S1, the meandering waveguide used in our experiment is fabricated on a 0.81 mm thick RO4003C substrate. In order to have the spin ensemble capable of interacting twice with the field, the microstrip with a width of 1.82 mm is connected with miter corners. All the angles of the bends of the miter corners in our device are designed to be 90°, which can maintain high transmission of the waveguide. The wavelength of the traveling photons in the waveguide is 2–4 cm in our probe frequency range (4–7 GHz). For the sake of illustrating the 'giant atom' physics, the distance between two coupling points of the outer (inner) YIG sphere is designed as 16.6 (8.3) cm, both larger than the wavelength of the microwave field (~4 cm). The 1 mm diameter yttrium-ion-garnet (YIG) spheres that we use are commercially available, and the [111] crystal axis of the YIG sphere is marked by a white stick glued to the sphere during the fabrication.

## Data availability

The data that support the finding of this study are available from the corresponding author upon reasonable request.

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

## Acknowledgements

This work has been supported by the National Natural Science Foundation of China (Grants Nos. 11934010, U1801661, 12174329, 11874249), Zhejiang Province Program for Science and Technology (Grant No. 2020C01019), and the Fundamental Research Funds for the Central Universities (No. 2021FZZX001-02).

## Author contributions

Z.Q.W. and Y.P.W. initiated the research project, Z.Q.W. designed the sample structure with input from Y.P.W. and J.L., Z.Q.W., J.G.Y., R.C.S., and W.J.W. realized the experiments and carried out the data analysis, Z.Q.W. and Y.P.W. developed the theory, Z.Q.W., Y.P.W., and J.Q.Y. drafted the manuscript, J.Q.Y. supervised the project, and all authors were involved in the discussion of results and the final manuscript editing.

## Competing interests

The authors declare no competing interests.
