## [Peer Review File · Nature Communications]

Reviewers' Comments:

Reviewer #1:

Remarks to the Author:

In this manuscript, the authors report the first experimental implementation of "giant atoms" using spin ensembles coupled to a waveguide. The usual situation in quantum optics is that atoms are much smaller than the wavelength of the light they interact with. It's only in the past decade that the possibility of (artificial) atoms interacting with light at multiple points, spaced wavelength distances apart, has been considered. There have been experimental demonstrations of such systems with superconducting qubits coupled to either microwave photons in meandering waveguides or surface acoustic waves, and proposals for some other implementations, but the present manuscript is to the best of my knowledge both the first proposal and the first experimental implementation with magnons in spin ensembles.

The experiments performed by the authors demonstrate thoroughly and convincingly how they can harness the interference effects due to the multiple coupling points of giant atoms. By tuning the resonance frequency of a spin ensemble using a bias magnetic field, the phase acquired when travelling between coupling points changes. This leads to either constructive or destructive interference, resulting in either high or low relaxation rate from the atom into the waveguide. The authors further demonstrate what happens when two giant atoms in a "nested" configuration (the two coupling points of one atom placed in between the two coupling points of the other atom) have their frequencies tuned. They find points where the coupling between the two atoms is purely dissipative and where it is purely coherent. In the latter case, the individual relaxation rate of one of the giant atoms also goes to zero.

The paper is quite well written, the figures are very nice, and the main text together with the supplementary material contains all necessary information to understand the experiments. The authors have demonstrated a new experimental platform to investigate the physics of giant atoms, which, as stated in the outlook, may offer some interesting possibilities compared to previous platforms with superconducting qubits. In light of this, I believe that the paper eventually can become suitable for publication in Nature Communications. However, before that, the authors need to properly discuss the relevant literature (see comment 1 below for details), since there is an important paper they didn't cite that already demonstrated some of the physics that the authors describe as new in the current manuscript. I also ask the authors to consider my other (minor) comments below.

1. The manuscript needs to properly discuss its results in relation to the work of Wen et al., Physical Review Letters 123, 233502 (2019). There, two superconducting qubits are placed close to the end of a transmission line. This corresponds to a setup with two small atoms in front of a mirror. As discussed in Ref. [5], this setup is essentially equivalent to the setup of two nested giant atoms studied in the current manuscript. The work of Wen et al. demonstrated a large coherent coupling between the two atoms, observable since the relaxation rate of one of them went to zero. This is essentially the same as the results discussed in the last paragraph on page 4 in the current manuscript. As noted in the text there, this effect would not be possible for small atoms in an open waveguide. However, small atoms in an open waveguide can demonstrate both purely coherent and purely dissipative coupling (see, e.g., Ref. [34]), while the current manuscript sometimes makes it sound as if this is something new for giant atoms.

2. In the introduction, the expression "giant physics" is used several times. I ask the authors to find a better wording in those places. Since nothing is said about atoms, light, or quantum optics in that expression, it could be misunderstood. For example, cosmology deals with giant length and time scales. Isn't that also "giant physics" then?

3. In the first paragraph of the second column on page 1, Ref. [20] (Vadiraj et al.) should be cited together with Ref. [8], not together with Refs. [16-22], since Ref. [20] is an actual experiment with a superconducting qubit, not a theoretical proposal.

4. At the start of the next paragraph, Ref. [5] should be cited together with Refs. [9, 18, 19], since it is the first study of interaction between giant atoms.

5. At the end of the same paragraph, the advantages of spin ensembles over superconducting qubits are discussed. However, not all of the things listed seem to be actual advantages. While the magnon mode frequency can be adjusted in situ through a bias magnetic field, so can the frequency of a superconducting qubit (see Ref. [8], for example). Even if the magnons have better coherence at room temperature, don't they need to be cooled down to millikelvin temperatures anyway for this experiment at GHz frequencies? Actually, I don't see any specification in the paper about what temperature the experiment was conducted at. This should be stated.

6. In Fig. 1b, it would be good to have a scale bar or something similar to show what the length scales in the experiment are.

7. Below Eq. (4), when stating all the fitted parameter values, it would be good to also give the value of v , which now only was given at the end of the supplementary material.

8. Some typos to correct:

* Page 1, second column, end of second paragraph: hybrid  hybrid

* In several places: times of π  multiples of π

* Several places in the supplementary material: lamb shift  Lamb shift

Reviewer #2:

Remarks to the Author:

The manuscript entitled 'Giant spin ensembles in waveguide magnonics' from Zi-Qi Wang et al. reports on their experimental observations of the interactions between two spatially separated YIG spheres, i.e., the giant spin ensembles termed by the authors, mediated by a meandering waveguide.

The authors develop a method that utilizes the magnetocrystalline anisotropy to locally tune one YIG's resonant frequency. Then, they design a meandering waveguide that can interact with each YIG twice to enable the self-interference of each YIG and the mediated coupling between two YIGs. They operate the microwave transmitting measurements at different frequencies to change the propagating phase delay between two YIGs. Consequently, they can control the self-interference of each YIG and switch the coupling type, coherent or dissipative, between two YIGs.

The paper is well written. The descriptions of both the experiment and the theoretical model are clear. The topic of the interactions between two resonant systems in a waveguide is important and of interest, because it may find applications in constructing quantum networks. Therefore, this topic attracts a lot of attention and has been studied in various physical systems, such as the superconducting qubit, acoustic systems and spin systems. However, I noticed that a similar work had been published in Physical Review Letters in this January [Phys. Rev. Lett. 128, 047701 (2022)], therefore I am nervous that this article written by Zi-Qi Wang et al. does not provide sufficient novel physics for publication in a high impact journal as Nature Communications.

In addition, I feel that some definitions and arguments in this manuscript may not be accurate. Please see the following comments:

1. By using a meandering waveguide, each YIG can interact with the waveguide twice. The authors call such YIG sphere the giant spin ensemble and emphasize the 'giant' physics. I'm not sure whether it is appropriate to rename a well-known system or not. I therefore would like the authors to provide further clarification.

2. The 'level attraction' shown in Fig. 4 (c) is so different from previous works [Phys. Rev. Lett. 128, 047701 (2022), or Ref.32]. Due to such difference, it seems that here in this work two resonances are only weakly coupled and so that only one resonance dip can be found near zero detunings. According to the authors' theoretical model, I agree that the dissipative coupling between two YIGs occurs. But it's too weak to produce a level attraction. Furthermore, the calculated results (dashed lines in Fig. 4 (c)) seem to be inconsistent with two modes' positions

from experiment as they approach each other.

3. The authors state that they observe an unexpected coherent coupling between two YIGs, even when the outer YIG is decoupled from the waveguide. I feel it's a little bit wired. If that, the radiative damping rate of the outer YIG, i.e., κ_o , is zero. According to Eq. (7), the coherent interaction strength is zero, which means that the coherent coupling doesn't exist. This paradox needs to be further discussed in their manuscript.

4. The coupling between two YIGs arises from their radiative damping rates into the waveguide. Considering that both YIGs have Gilbert damping rates, it is very hard to tune this coupled system into a strong coupling regime. I notice that the authors give a strong criterion at the end of the discussion of the experiment. It may need to be clearly proved.

I believe that this work presents some interesting ideas. But with the previous comments in mind, the above questions need to be addressed to convince the readers that a sufficient level of progress/novelty was achieved for publication in Nature Communications.

Reviewer #3:

Remarks to the Author:

The authors have investigated 'giant' physics by exploiting ferromagnetic resonance modes of yttrium iron garnet (YIG) at room temperature in a meandering waveguide. The subject is quite interesting in the broad domain of light-matter interactions as well as magnonics. The authors have conducted detailed and precise experiments for the demonstrations. The data are supported by theoretical models. The figures are well-organized. The conclusions are supported by the results.

The novelty of the manuscript includes a new platform to study light-matter interaction at room temperature. Furthermore, this architecture offers ease of tunability/control as a ferromagnetic spin ensemble is used. It will potentially attract wide attention across domains. Therefore, I feel that this work is suitable for publication in Nature Communications. Nevertheless, the authors may want to address the following suggestions.

- It will be good for the readers to explicitly describe the sample details (YIG fabrication details, size etc.).
- Similarly, include the waveguide details (fabrication details, size etc.).
- Please comment on the choice for the cavity separation (L). Also, discuss the dependence of the waveguide design on the results.
- The author may want to add a discussion on how to improve the effect based on their experience for future advancement.
- It will be good to include an experimental image of the sample along with the meandering waveguide.
- Observation of anti-crossing of magnon modes has been observed in the past for different contexts. Can the author comment on such observations and distinguish their results?
- Include the experimental details such as the power used for the RF current.
- Bias current has been used for the plots throughout the manuscript and it is misleading as it is used to drive an electromagnet. It is the bias field that is more appropriate. The authors may revise the relevant figures accordingly.
- The Kittel mode has been investigated in this report for the demonstrations. However, it will be good to add a discussion on the variation of the higher order mode between the coupling state and the un-coupled state.
- Comment on how to miniaturize such a sample design.
- Add a discussion by comparing the results with previous demonstrations (superconducting qubit) that are already cited in the manuscript.

Reply to Referee #1's comments

First of all, we would like to thank referee #1 for his/her quick, positive and constructive comments. Below we respond to them one by one.

Comment:

In this manuscript, the authors report the first experimental implementation of "giant atoms" using spin ensembles coupled to a waveguide. The usual situation in quantum optics is that atoms are much smaller than the wavelength of the light they interact with. It's only in the past decade that the possibility of (artificial) atoms interacting with light at multiple points, spaced wavelength distances apart, has been considered. There have been experimental demonstrations of such systems with superconducting qubits coupled to either microwave photons in meandering waveguides or surface acoustic waves, and proposals for some other implementations, but the present manuscript is to the best of my knowledge both the first proposal and the first experimental implementation with magnons in spin ensembles.

Reply:

We fully agree with referee #1's description of our manuscript and also appreciate his/her review on the state of the art of the giant atom physics.

Comment:

The experiments performed by the authors demonstrate thoroughly and convincingly how they can harness the interference effects due to the multiple coupling points of giant atoms. By tuning the resonance frequency of a spin ensemble using a bias magnetic field, the phase acquired when travelling between coupling points changes. This leads to either constructive or destructive interference, resulting in either high or low relaxation rate from the atom into the waveguide. The authors further demonstrate what happens when two giant atoms in a "nested" configuration (the two coupling points of one atom placed in between the two coupling points of the other atom) have their frequencies tuned. They find points where the coupling between the two atoms is purely dissipative and where it is purely coherent. In the latter case, the individual relaxation rate of one of the giant atoms also goes to zero.

Reply:

We thank referee #1 for this positive comment and agree with his/her comprehensive summary of our work.

Comment:

The paper is quite well written, the figures are very nice, and the main text together with the supplementary material contains all necessary information to understand the experiments. The authors have demonstrated a new experimental platform to investigate the physics of giant atoms, which, as stated in the outlook, may offer some interesting possibilities compared to previous platforms with superconducting qubits. In light of this, I believe that the paper eventually can become suitable for publication in Nature Communications.

Reply:

We thank referee #1 for this positive comment.

Comment:

However, before that, the authors need to properly discuss the relevant literature (see comment 1 below for details), since there is an important paper they didn't cite that already demonstrated some of the physics that the authors describe as new in the current manuscript. I also ask the authors to consider my other (minor) comments below.

Reply:

We greatly appreciate referee #1 for the constructive comments and helpful suggestions. We have fully followed these comments and suggestions to revise our manuscript. Below we respond to the comments and suggestions one by one.

Comment:

1. The manuscript needs to properly discuss its results in relation to the work of Wen et al., Physical Review Letters 123, 233502 (2019). There, two superconducting qubits are placed close to the end of a transmission line. This corresponds to a setup with two small atoms in front of a mirror. As discussed in Ref. [5], this setup is essentially equivalent to the setup of two nested giant atoms studied in the current manuscript. The work of Wen et al. demonstrated a large coherent coupling between the two atoms, observable since the relaxation rate of one of them went to zero. This is essentially the same as the results discussed in the last paragraph on page 4 in the current manuscript. As noted in the text there, this effect would not be possible for small atoms in an open waveguide. However, small atoms in an open waveguide can demonstrate both purely coherent and purely dissipative coupling (see, e.g., Ref. [34]), while the current manuscript sometimes makes it sound as if this is something new for giant atoms.

Reply:

We thank referee #1 for this insightful comment. He/she pointed out two relevant and interesting works: Wen, P. et al., Physical Review Letters 123, 233502 (2019); Van Loo, A. F. et al., Science 342, 1494-1496 (2013).

The work of Wen et al. demonstrates the collective behavior of two superconducting qubits (*small atoms*) placed close to the end of a transmission line. In this study, the experimental results and effective Hamiltonian of the system are equivalent to the nested *giant atoms*. In the revised manuscript, we have appropriately cited this fascinating work (see new Ref. [33] cited in the revised manuscript). Correspondingly, we have added a sentence in the right column of page 1 as follows: **“This phenomenon was previously observed with two superconducting qubits (small atoms) placed close to the end of a transmission line.”**

It should be noted that in the two-port waveguide setup, the situation is different. As demonstrated by Van Loo et al. (i.e., Ref. [38] in the revised manuscript), the interaction between two superconducting qubits (*small atoms*) is mediated by the standard two-port waveguide. The coupling (either coherent or dissipative) between these two small atoms can be regulated by the

resonance frequency, but if one of the two small atoms is decoupled from the waveguide, there is no longer any interaction between these two small atoms. However, a novel phenomenon can occur in the case of giant atoms. Our work highlights the counter-intuitive phenomenon that even the outer giant spin ensemble is decoupled from the two-port waveguide, we can still observe the coherent interaction between the two giant spin ensembles. This phenomenon is incapable with two small atoms in the traditional two-port waveguide configuration, as established in Ref. [38].

Nevertheless, referring to the work by Wen et al. (Ref. [33]), it is indeed possible to construct coherent interaction between small atoms when one of the atoms is decoupled from the environment in the configuration of transmission line with an end. In order to avoid any misunderstanding, we have thoroughly revised the text relevant to all potentially misleading descriptions. For example, on page 5, we have added the phrase “**two-port waveguide**” while discussing and comparing with the waveguide-mediated interaction between small atoms studied by Van Loo et al. (Ref. [38]). Also, we have added an explanation in the last paragraph of page 1 as follows: “**In contrast, the coupled giant spin ensembles are implemented in the two-port waveguide system**”.

In addition, we would like to underline that the waveguide configuration associated with ‘giant atom’ physics may have applications in the construction of a quantum network, as it is based on a two-port setup that extends in both two directions, in contrast to the transmission line with an end.

Comment:

2. In the introduction, the expression "giant physics" is used several times. I ask the authors to find a better wording in those places. Since nothing is said about atoms, light, or quantum optics in that expression, it could be misunderstood. For example, cosmology deals with giant length and time scales. Isn't that also "giant physics" then?

Reply:

We thank referee #1 for this helpful comment. To avoid misleading, we have modified the expression to ‘**giant atom’ physics** in the revised manuscript.

Comment:

3. In the first paragraph of the second column on page 1, Ref. [20] (Vadiraj et al.) should be cited together with Ref. [8], not together with Refs. [16-22], since Ref. [20] is an actual experiment with a superconducting qubit, not a theoretical proposal.

Reply:

We thank referee #1 for pointing out this important information. In the revised manuscript, the work by Vadiraj et al. is cited together with Ref. [8].

Comment:

4. At the start of the next paragraph, Ref. [5] should be cited together with Refs. [9, 18, 19], since it is the first study of interaction between giant atoms.

Reply:

We thank referee #1 for this helpful comment. In the revised manuscript, Ref. [5] is cited together with Refs. [9,19,20].

Comment:

5. At the end of the same paragraph, the advantages of spin ensembles over superconducting qubits are discussed. However, not all of the things listed seem to be actual advantages. While the magnon mode frequency can be adjusted in situ through a bias magnetic field, so can the frequency of a superconducting qubit (see Ref. [8], for example). Even if the magnons have better coherence at room temperature, don't they need to be cooled down to millikelvin temperatures anyway for this experiment at GHz frequencies? Actually, I don't see any specification in the paper about what temperature the experiment was conducted at. This should be stated.

Reply:

We thank referee #1 for this thoughtful question. Compared with the superconducting qubits, the advantage of the flexibility and adjustability of the magnon mode is mainly manifested in its large continuously tunable frequency range. The magnon mode frequency can be tuned from a few hundred megahertz to several tens of gigahertz by manipulate the bias magnetic field, which is hard to realize with superconducting qubits. Our work also utilizes the magnetocrystalline anisotropy field to tune the frequency of magnon mode by rotating the YIG sphere in a static magnetic field, which is favorable for individually tuning the magnon mode frequency in different YIG spheres. However, in order to avoid overemphasizing the advantages of spin ensembles while ignoring the unique advantages of superconducting qubits in demonstrating the physics of giant atoms, we have deleted "**rather than a superconducting qubit**". Correspondingly, we have modified the description of the advantage as follows: "**The benefits of choosing a ferromagnetic spin ensemble include the flexible and wide-range adjustability of magnon mode frequency (a few hundred megahertz to several tens of gigahertz), ...**".

Actually, our experiment was conducted at room temperature. The expression "favorable magnon coherence at room temperature" in this paragraph is intended to illustrate the low dissipation rate of the magnon mode at room temperature. Since the Curie temperature of yttrium iron garnet is as high as 550 K, the ferromagnetic spin ensemble can be used to demonstrate the self-interference effect related to the giant atom physics at room temperature. To make the statement clear, we have revised the expression as follows: "**...low dissipation rate at room temperature...**" and added the specification of the experimental temperature in the **right column on page 2**.

Comment:

6. In Fig. 1b, it would be good to have a scale bar or something similar to show what the length scales in the experiment are.

Reply:

We thank referee #1 for this helpful comment. We have added a scale bar below Fig. 1b.

Comment:

7. Below Eq. (4), when stating all the fitted parameter values, it would be good to also give the value of v , which now only was given at the end of the supplementary material.

Reply:

We thank referee #1 for this helpful comment. We have added the fitted value of v in the paragraph below Eq. (4).

Comment:

8. Some typos to correct:

* Page 1, second column, end of second paragraph: *hybird*  *hybrid*

* In several places: *times of π*  *multiples of π*

* Several places in the supplementary material: *lamb shift*  *Lamb shift*

Reply:

We thank referee #1 for carefully reading our manuscript. We have corrected these typos in the main text and supplementary materials.

Reply to Referee #2's comments

First of all, we would like to thank referee #2 for his/her quick, positive and constructive comments. Below we respond to them one by one.

Comment:

The manuscript entitled 'Giant spin ensembles in waveguide magnonics' from Zi-Qi Wang et al. reports on their experimental observations of the interactions between two spatially separated YIG spheres, i.e., the giant spin ensembles termed by the authors, mediated by a meandering waveguide.

The authors develop a method that utilizes the magnetocrystalline anisotropy to locally tune one YIG's resonant frequency. Then, they design a meandering waveguide that can interact with each YIG twice to enable the self-interference of each YIG and the mediated coupling between two YIGs. They operate the microwave transmitting measurements at different frequencies to change the propagating phase delay between two YIGs. Consequently, they can control the self-interference of each YIG and switch the coupling type, coherent or dissipative, between two YIGs.

Reply:

We agree with referee #1's summary regarding our manuscript.

Comment:

The paper is well written. The descriptions of both the experiment and the theoretical model are clear. The topic of the interactions between two resonant systems in a waveguide is important and

of interest, because it may find applications in constructing quantum networks. Therefore, this topic attracts a lot of attention and has been studied in various physical systems, such as the superconducting qubit, acoustic systems and spin systems.

Reply:

We thank referee #2 for this positive comment.

Comment:

However, I noticed that a similar work had been published in Physical Review Letters in this January [Phys. Rev. Lett. 128, 047701 (2022)], therefore I am nervous that this article written by Zi-Qi Wang et al. does not provide sufficient novel physics for publication in a high impact journal as Nature Communications.

Reply:

We thank referee #2 for evaluating our manuscript and bringing our attention to the interesting work of Yi Li et al., Phys. Rev. Lett. 128, 047701 (2022). We are sorry for not noticing this work. In the revised manuscript, we have cited it as Ref. [31].

After carefully reading the PRL work, we see the difference and disparity between our research goal and that of the PRL work. The objective of this PRL work is to investigate the long-range interactions between spin ensembles mediated by superconducting circuits, which does not involve the specific structure and physics of giant atoms. However, our primary objective is to illustrate the ‘giant atom’ physics in spin ensemble systems, exploiting the rich interference effect of ‘giant atom’ physics to manipulate the interaction between the spin ensemble and its environment, as well as the interaction between spin ensembles. Our central results involve a new light-matter interaction beyond the dipole approximation, which is not studied in the PRL work above. In this regard, we are convinced that our experimental demonstrations are sufficiently novel. Below we would like to describe in detail the primary novel aspects of our work. We sincerely hope that referee #2 can find these explanations useful to see the novelty of our work.

In the small spin ensemble case, the YIG sphere is treated as a point in the light-matter interaction. For example, in the PRL work of Yi Li et al., the size of the YIG sphere ($\sim 250 \mu\text{m}$) is much smaller than the wavelength of the microwave field (on the order of a centimeter) with which it interacts, as shown in Figure R1a below. However, in our work, the size of the YIG sphere is also small, but its effective size is effectively enlarged by using the meandering waveguide, which is also on the order of a centimeter. This non-negligible effective size of the YIG sphere ($\sim \text{cm}$) in the light-matter interaction leads to the self-interference effect between the light-matter coupling points. This effect is well reflected in the frequency-dependent relaxation rate of the magnon mode in the YIG sphere, and the periodic decoupling of the single spin ensemble from the environment (waveguide) is observed, as depicted in Figs. 2a and 2b of the main text. This is an essential feature of ‘giant atom’ physics that cannot be realized with small atoms and small spin ensembles. This enables us to realize the switching between decoupling and coupling by simply adjusting the frequency.

We further demonstrate the intriguing collective behavior of two nested giant spin ensembles. In the coupled giant atoms system, the different topologies of the coupling points can give rise to various interaction behaviors. A representative example is the decoherence-free interaction in braided structure depicted in Fig. R1c, which is examined experimentally using superconducting qubits [Kannan et al., Nature 583, 775-779 (2020)]. As depicted in Fig. R1b, our work investigates experimentally for the first time the nested configuration. In this setup, we find that the coupling between giant spin ensembles mediated by meandering waveguide and the coupling between small spin ensembles mediated by conventional waveguide [as demonstrated in Yi Li et al., Phys. Rev. Lett. 128, 047701 (2022)] are very different. Even when the outer giant spin ensemble is decoupled from the waveguide, a level repulsion is observed. In contrast, in the small spin ensemble system, when one of the spin ensembles is decoupled from the waveguide, the interaction between the spin ensembles becomes unavailable. This novel phenomenon in our system of giant spin ensembles can stimulate further investigations. Moreover, in the following responses to specific questions, we would like to explain in detail that in the nested giant atom system, it is feasible to achieve strong coherent interactions through the mediation of open waveguides. However, in small atom systems, strong coherent coupling mediated by a waveguide is unachievable. This unique effect will enable one to control the interactions between nodes in future quantum networks in a flexible and efficient manner.

Figure R1. Configurations of the waveguide mediated coupling between (a) two small spin ensembles, (b) the nested giant spin ensembles, (c) the braided giant spin ensembles.

Finally, we would like to mention that, to the best of our knowledge, this is the first experimental realization of the nested structure of giant atoms/spin ensembles. Since the rapid theoretical development of the ‘giant atom’ physics (see Refs. [17-22] in the main text), only superconducting circuits have been employed to demonstrate novel light-matter interaction beyond the dipole approximation. Our demonstration has timely provided a new and easy-to-operate platform to investigate the versatile effects of ‘giant atom’ physics. This novelty is acknowledged by referee #1’s comment: “...but the present manuscript is to the best of my knowledge both the first proposal and the first experimental implementation with magnons in spin ensembles.”

Through the detailed explanations above, we hope that referee #2 finds our work to be original in exploring the physics of ‘giant atoms’ by using unique configurations of spin ensembles interacting with a waveguide.

Comment:

In addition, I feel that some definitions and arguments in this manuscript may not be accurate. Please see the following comments:

Reply:

We thank referee #2 for helpful and constructive comments. We have studied the rest of referee #2's comments and addressed them accordingly. Below we respond to them one by one.

Comment:

1. By using a meandering waveguide, each YIG can interact with the waveguide twice. The authors call such YIG sphere the giant spin ensemble and emphasize the 'giant' physics. I'm not sure whether it is appropriate to rename a well-known system or not. I therefore would like the authors to provide further clarification.

Reply:

We thank referee #2 for this helpful comment. According to the original Kockum's study (Ref. [3] in the main text), when the size of the atoms becomes comparable to the wavelength of the field with which they interact, the dipole approximation no longer applies and the atom is referred to as a 'giant atom'. The giant atom leads to a series of striking effects incapable with the small atoms (Refs. [17-22]). The non-local interaction between the atom and the field is the basis of 'giant atom' physics (**depicted in Fig. R2**). We use the meandering waveguide to have the spin ensemble interact several times with the field, and this scheme is purposed by Kockum (Ref. [3]) and implemented for the first time by Kannan in a superconducting circuit (Ref. [5]). This concept prompted us to explore the physics of 'giant atoms' on a novel platform using YIG spheres. The structure of the meandering waveguide enables the traveling photons interact twice with the magnon mode in the spin ensemble, resulting in the effective enlargement of the spin ensembles. As also noted by referee #1, the expression 'giant physics' is certainly misleading. Therefore, we have modified the 'giant physics' to 'giant atom' physics in the revised manuscript.

Figure R2. Illustrations of small atom (**left**), and giant atom (**right**).

Comment:

2. The 'level attraction' shown in Fig. 4 (c) is so different from previous works [Phys. Rev. Lett. 128, 047701 (2022), or Ref.32]. Due to such difference, it seems that here in this work two resonances are only weakly coupled and so that only one resonance dip can be found near zero

detunings. According to the authors' theoretical model, I agree that the dissipative coupling between two YIGs occurs. But it's too weak to produce a level attraction.

Reply:

We thank referee #2 for this insightful comment. Because of the difference in relative magnitude between the strength of the dissipative coupling and the dissipation rates of the coupled modes, the 'level attraction' depicted in Figure 4c appears different from the previous work. In other words, this is because they are in different dissipative coupling regimes.

First, let us briefly review the various coherent-coupling regimes. According to the definition in the work of X. Zhang et al., Phys. Rev. Lett. 113, 156401 (2014), in a two-mode coherently coupled system (A mode and B mode), by comparing the coherent coupling strength (g_{ab}) and the dissipation rate of each mode (κ_a, κ_b), the coupled system can be divided into weak coupling region ($g_{ab} < \{\kappa_a, \kappa_b\}$), strong coupling region ($g_{ab} > \kappa_a, \kappa_b$), magnetically induced transparency (MIT) region ($\kappa_b < g_{ab} < \kappa_a$), and Purcell effect region ($\kappa_a < g_{ab} < \kappa_b$). In different coupling regions, different spectral mappings can be obtained, as shown in Fig. R3 (c) and (e) [adapted from Phys. Rev. Lett. 113, 156401 (2014)] corresponding to the magnetically induced transparency regime and Purcell effect regime, respectively. Despite the diverse appearances of the spectral mappings, the spectral features are always characterized by energy level repulsion, which is the manifestation of coherent coupling.

Figure R3. Different coupling regimes of coherently coupled system. (c) Spectral mapping obtained in the magnetically induced transparency regime; (d) Spectral mapping obtained in the Purcell effect regime [adapted from Phys. Rev. Lett. 113, 156401 (2014)].

It is shown in Fig. R4a that the 'level attraction' is induced by the dissipative coupling between two resonators, in which the resonators cooperatively dissipate into the open environment. Similarly, regimes of dissipative coupling in a two-mode system can be classified. The magnitude of the dissipative coupling strength is equal to the square root of the product of two external dissipation rates of the resonators $\Gamma = \sqrt{\kappa\gamma}$, and each resonator has its own intrinsic dissipation rate (α and β). Therefore, the quantities to be compared are the strength of the dissipative

coupling $\Gamma = \sqrt{\kappa\gamma}$ and the total dissipation rates of the two resonators $\kappa + \alpha$ and $\gamma + \beta$. Obviously, in a dissipative coupling system, the dissipative coupling strength cannot exceed the total dissipation rates of the two resonators at the same time ($\Gamma = \sqrt{\kappa\gamma} > \{\kappa + \alpha, \gamma + \beta\}$), i.e., the dissipative coupling system cannot reach the strong coupling regime. However, there are circumstances in which the external dissipation rate of one of the resonators is significantly greater than the external dissipation rate of the other resonator and the intrinsic dissipation rates of the two resonators are relatively small. In this case, the magnetically induced transparency regime ($\kappa + \alpha > \sqrt{\kappa\gamma} > \gamma + \beta$) and Purcell effect regime ($\gamma + \beta > \sqrt{\kappa\gamma} > \kappa + \alpha$) can be achieved. In more cases, one can only achieve weak coupling regime of dissipative coupling, that is, when the two resonators are relatively equivalent ($\kappa \approx \gamma, \alpha \approx \beta, \sqrt{\kappa\gamma} < \{\kappa + \alpha, \gamma + \beta\}$).

As illustrated in Fig. R4b, we distinguish the different dissipative coupling regimes by comparing the relative magnitudes of the dissipative coupling strength and the damping rates of two resonators. The previous work [M. Harder et al., Phys. Rev. Lett. 121, 137203 (2018)] belongs to the magnetically induced transparency (MIT) regime, and our work is in the weak coupling regime. In the work of M. Harder et al., the ‘level attraction’ occurs between the cavity mode and the magnon mode, where the cavity mode’s external damping rate is two orders of magnitude greater than the magnon mode’s external damping rate, so the system is in the MIT regime. Using the same parameters in M. Harder’s PRL work, the transmission mapping can be calculated and plotted in Fig. R4c. It shows the typical ‘level attraction’ appearance. The MIT window appears and passes through the broad cavity resonance. The transmission spectrum at resonance is shown in Fig. R4e and an MIT peak is clearly visible. The result is consistent with the experimental result demonstrated in the PRL work of M. Harder.

Figure R4. **a.** Schematic of the dissipative coupling mechanism; **b.** Different coupling regimes of the dissipatively coupled resonators; **c.** Level attraction in the magnetic induced transparency regime of the dissipative coupling; **d.** Level attraction-like effect in the weak coupling regime of the dissipative coupling; **e.** and **f.** Transmission spectra corresponding to the resonance positions in **c** and **d**, respectively.

In our work, dissipative coupling occurs between two giant spin ensembles with comparable damping rates, indicating that the system is in the weak coupling regime. Utilizing the same parameters as in our work, the transmission mapping is plotted in Fig. R4d. We can find that two

modes are degenerate at resonance, and the transmission spectrum exhibits a Lorentzian broadened line shape, as depicted in Fig. R4f. Although it is not as typical as the instance observed in the MIT regime, the spectral energy level attraction also reflects the dissipative coupling. In order to distinguish the energy level attractions observed in the previous work and in our present work, we modified the phrase ‘level attraction’ as ‘level attraction-like’ in the revised manuscript.

In addition, as indicated by referee #2, the level attraction observed in the other paper [Phys. Rev. Lett. 128, 047701 (2022)] also appears distinct from ours. We think that the situation in this PRL work is also essentially in the weak coupling regime, since the dissipative interaction is demonstrated between two nearly identical YIG spheres. Based on the transmission mapping, we infer that the difference may stem from higher-order magnetostatic modes that are involved in the dissipative coupling. Near the resonance, the spectral lines of other magnetostatic modes can be seen in this PRL work. However, only Kittel modes are involved in our work, without contributions from the higher-order magnetostatic modes.

Since the strong coupling region of dissipative coupling cannot be reached with the current technology, and the magnetically induced transparency regime has been commonly referred to as energy level attraction, we refer to the spectral lines with the attractive characteristics of dissipative coupling in other regimes as energy level attraction-like. Based on referee #2's suggestion, we have modified the manuscript accordingly.

Comment:

Furthermore, the calculated results (dashed lines in Fig. 4 (c)) seem to be inconsistent with two modes' positions from experiment as they approach each other.

Reply:

We thank referee #2 for the helpful comment. According to the referee suggestion, we have revised the calculated result (dashed line) in **Fig. 4c** of the main text.

Comment:

3. The authors state that they observe an unexpected coherent coupling between two YIGs, even when the outer YIG is decoupled from the waveguide. I feel it's a little bit wired. If that, the radiative damping rate of the outer YIG, i.e., κ_o , is zero. According to Eq. (7), the coherent interaction strength is zero, which means that the coherent coupling doesn't exist. This paradox needs to be further discussed in their manuscript.

Reply:

We thank referee #2 for this helpful comment. Equations (4) and (7) in the main text describe the phase dependence of the radiative decay rate and the coherent coupling strength, respectively. It should be pointed that the κ_o is defined as the radiative damping rate of the outer giant spin ensemble at the coupling points, which is fitted to be 0.70 MHz. Equation (4) gives the radiative damping rate of the giant spin ensemble $\kappa_{o,G}$, which is zero in the case of purely coherent coupling, where the subscript G denotes ‘giant’.

Intuitively, we think that the two giant spin ensembles will not interact with each other since the outer giant spin ensemble is decoupled from the waveguide. Nevertheless, Eq. (7) and Fig. 3b reveal a non-zero coherent coupling between two giant spin ensembles, which is clearly shown in Fig. 4g. This unexpected novel phenomenon is explained in the right column of the main text on page 4.

We recognize that the subscripts of κ_o and $\kappa_{o,G}$ may lead to some misunderstandings. In the revised manuscript, we have elaborated on these parameters at the given places.

Comment:

4. The coupling between two YIGs arises from their radiative damping rates into the waveguide. Considering that both YIGs have Gilbert damping rates, it is very hard to tune this coupled system into a strong coupling regime. I notice that the authors give a strong criterion at the end of the discussion of the experiment. It may need to be clearly proved.

Reply:

We thank referee #2 for this insightful comment. In our work, the damping rates of the nested two giant spin ensembles consists of the intrinsic damping rates β_i, β_o and radiative damping rates $\kappa_{i,G}, \kappa_{o,G}$. The Gilbert damping rate is included in the intrinsic damping rate. It should be noted that each giant spin ensemble is here designed to have two coupling points with the meandering waveguide. The radiative damping rates at the couplings points of the inner and outer giant spin ensembles are κ_i and κ_o , respectively. According to Eq. (7), it is obvious that the maximal value of the coherent coupling strength is proportional to the product of the radiative damping rates of the inner and outer giant spin ensembles at the coupling points ($J_{max} \propto \sqrt{\kappa_i \kappa_o}$). Despite the outer giant spin ensemble being decoupled from the waveguide (i.e., $\kappa_{o,G} = 0$), the coherent coupling strength still exists, as mentioned above. Utilizing this advantage, if the radiative damping rate of the outer resonator κ_o at the coupling point is large enough (like the superconducting qubits), then we can have $\sqrt{\kappa_i \kappa_o} > \max\{\kappa_{i,G} + \beta_i, \beta_o\}$, and the system will enter the strong coupling regime in the open environment.

Comment:

I believe that this work presents some interesting ideas. But with the previous comments in mind, the above questions need to be addressed to convince the readers that a sufficient level of progress/novelty was achieved for publication in Nature Communications.

Reply:

We are very grateful to referee #2 for providing us with many valuable and insightful suggestions and comments. We fully adopted the comments and revised the unclear parts of the manuscript, so as to eliminate the misunderstandings. Also, we have carefully explained the important achievements and novelties of our manuscript regarding the demonstration of the intriguing ‘giant atom’ physics in the new platform of spin ensemble system. We sincerely hope that our answers can be satisfactory to referee #2.

Reply to Referee #3's comments

First of all, we would like to thank referee #3 for his/her quick, positive and constructive comments.

Comment:

The authors have investigated 'giant' physics by exploiting ferromagnetic resonance modes of yttrium iron garnet (YIG) at room temperature in a meandering waveguide. The subject is quite interesting in the broad domain of light-matter interactions as well as magnonics. The authors have conducted detailed and precise experiments for the demonstrations. The data are supported by theoretical models. The figures are well-organized. The conclusions are supported by the results.

Reply:

We thank referee #3 for this positive comment and agree with the description of our work.

Comment:

The novelty of the manuscript includes a new platform to study light-matter interaction at room temperature. Furthermore, this architecture offers ease of tunability/control as a ferromagnetic spin ensemble is used. It will potentially attract wide attention across domains. Therefore, I feel that this work is suitable for publication in Nature Communications. Nevertheless, the authors may want to address the following suggestions.

Reply:

We thank referee #3 for the positive comments. The constructive suggestions are very beneficial for improving the manuscript. Following these, we have revised the manuscript accordingly. Below are our one by one responses.

Comment:

•It will be good for the readers to explicitly describe the sample details (YIG fabrication details, size etc.).

Reply:

We thank referee #3 for this helpful suggestion. We have added a subsection entitled 'Device design' in the supplementary materials, regarding the YIG sample and the details of the meandering waveguide device design.

The YIG sphere is commercially available. We have included the purchase information in the main text as follows: “**The spin ensembles we use are 1 mm-diameter single-crystal yttrium iron garnet (YIG) spheres, which are commercially available and manufactured by Ferrisphere Inc. [34]**”. In references, Ref. [34] is indicated as <http://www.ferrisphere.com>.

Comment:

•Similarly, include the waveguide details (fabrication details, size etc.).

Reply:

We thank referee #3 for this helpful suggestion. We have added the design and fabrication details of the meandering waveguide in the 'Device design' subsection of the supplementary materials, including a photograph of the device and details of the design and the materials (**Rogers RO4003C substrate**) used for fabricating the meandering waveguide.

Comment:

•Please comment on the choice for the cavity separation (L). Also, discuss the dependence of the waveguide design on the results.

Reply:

We thank referee #3 for this helpful suggestion. In order to demonstrate the 'giant atom' physics, the separation of the coupling points we choose depends on the wavelength of the field that interacts with the spin ensembles. In our work, the effective wavelength of the microwave field is 2~4 cm in our experimental frequency range (4~7 GHz). The length of L we choose for the inner and outer giant spin ensembles are 8.3 cm and 16.6 cm, respectively, both larger than 4 cm. In this case, the two spin ensembles can both be treated as 'giant' spin ensembles. According to the suggestion of referee #3, we have added the discussion of the choice for the separation (L) in the right column of page 2 as follows: **"To illustrate the 'giant atom' physics, the distance between the two coupling points of the inner (outer) sphere is designed to be 8.3 (16.6) cm, all greater than the wavelength (about 2-4 cm) of the microwave they interact with."**

Comment:

•The author may want to add a discussion on how to improve the effect based on their experience for future advancement.

Reply:

We thank referee #3 for this helpful suggestion. We expect that a variety of novel phenomena beyond the dipole approximation can be implemented using the new platform provided in this manuscript. Following referee #3's suggestion, we have added three paragraphs in the section of Conclusion and Outlook on page 6 to discuss the novel microwave and optical devices base on the 'giant atom' physics, the building of 'giant atom' networks, and the quantum effects demonstration (see below).

"From previous studies and ours, we can expect the future development of 'giant atom' physics in the following three aspects. First, it is expected to construct more novel microwave and optical devices base on the 'giant atom' physics. In the 'giant atom' system, the self-interference effect can be utilized and manipulated by tuning the phase difference between the coupling points, which gives rise to the periodic coupling and decoupling of the resonators or emitters. The phase difference can be controlled by the resonance frequency of the resonators or emitters coupled to the meandering waveguide. When decoupling occurs, the transmission spectrum reveals maximum transmission, whereas the spectrum reveals very low transmission and considerably high reflection when the coupling is large. Therefore, we obtain a band-pass filter that is frequency-dependent

and tunable. With the self-interference effect of 'giant atom' physics, it is promising to design more microwave and optical devices with a variety of functions.

The second aspect is related to the building of 'giant atom' networks. Beyond the dipole approximation, the self-interference effect can result in the realization of periodic coupling and decoupling, and the coupling between different giant atoms can also be regulated and become non-local. Using these features, we may construct a meandering waveguide-mediated network of giant atoms to realize a variety of information processing and storage schemes.

The third one is about demonstrating quantum effects in the 'giant atom' system. The phenomena addressed in this work only exploit the interference effect in a light-matter interaction beyond the dipole approximation, which has correspondences in both classical and quantum physics. For pure quantum effects, such as the quantum entanglement, vacuum squeezed state, etc., particularly the non-local effect of the quantum entanglement, one can expect to perform some interesting demonstrations in the context of 'giant atom' physics in the future."

Comment:

•It will be good to include an experimental image of the sample along with the meandering waveguide.

Reply:

We thank referee #3 for this helpful suggestion. We have added a photograph of the device (**Fig. S1**) along with the meandering waveguide in the 'Device design' subsection of the supplementary materials.

Comment:

• Observation of anti-crossing of magnon modes has been observed in the past for different contexts. Can the author comment on such observations and distinguish their results?

Reply:

We thank referee #3 for this helpful question. In the cavity QED system, it is relatively easy to observe the anti-crossing because a strong coupling is easily achievable. When the YIG sphere is placed at the antinode of the cavity-mode magnetic field, if the quality factor of the cavity is relatively high, the coupled system can easily enter the **strong** coupling regime, that is, the coherent coupling strength is larger than the linewidths of both the cavity mode and the magnon mode. The direct magnetic-dipole interaction between the cavity mode and the magnon mode is reflected as the anti-crossing (energy level repulsion) in the spectra measurement (see, e.g., Refs. [25,26] in the revised manuscript).

Our experiment, in contrast, is demonstrated using the waveguide QED system. In a normal open waveguide system, coherent coupling mediated by traveling-wave photons is feasible, but **strong** coherent coupling is **challenging** since the coupling is mediated by a dissipative process. It is difficult for the coherent coupling strength to exceed the linewidth. Therefore, in a normal

waveguide QED system, it is difficult to observe anti-crossing (Ref. [38]). Nevertheless, as demonstrated in the nested giant atom structure in our experiment, the dissipation of the giant spin ensemble can become zero simultaneously with the occurrence of coherent coupling. In this case, the anti-crossing resulting from the coherent coupling is evident, as shown in **Figure 4g** of the manuscript.

According to referee #3's suggestion, we have emphasized this point in the revised manuscript.

Comment:

•Include the experimental details such as the power used for the RF current.

Reply:

We thank referee #3 for this helpful suggestion. We have added the details of the probe signal power on page 2 as follows: “...and the power of the probe signal is -10 dBm...”.

Comment:

•Bias current has been used for the plots throughout the manuscript and it is misleading as it is used to drive an electromagnet. It is the bias field that is more appropriate. The authors may revise the relevant figures accordingly.

Reply:

We thank referee #3 for this helpful suggestion. We have replaced the bias current with the bias field in the revised manuscript and modified the corresponding figures (**Fig. 1e, Fig. 2a, b**).

Comment:

•The Kittel mode has been investigated in this report for the demonstrations. However, it will be good to add a discussion on the variation of the higher order mode between the coupling state and the un-coupled state.

Reply:

We thank referee #3 for this helpful suggestion. The variation of the higher-order magnetostatic mode between the coupling state and the un-coupled state exists. However, due to a small magnetic-dipole moment of the higher-order mode, the coupling between the higher-order mode and the waveguide is much weaker. In our work, the signal of the higher-order mode is too small to be clearly identified, as shown in Fig. 2b of the main text and in the red boxes of figure R5 (see below).

Figure R5. Reprinted Figure 2 of the main text. The red boxes label the higher-order modes, which are hard to be clearly distinguished due to the weak coupling between the higher-order mode and the waveguide.

Indeed, harnessing higher-order modes can increase the control degrees of freedom of the giant spin ensemble system and may stimulate exploration of more phenomena in the future. Thus, as suggested by referee #3, we have added the following discussion in the section of Conclusion and Outlook (see the last paragraph) as follows: **“In addition, the higher-order magnetostatic modes that are not discussed in the present work may be of interest in future study. We can harness these higher-order modes to increase the control degrees of freedom of the system and explore exotic phenomena in the multi-mode ‘giant atom’ system.”**

Comment:

•Comment on how to miniaturize such a sample design.

Reply:

We thank referee #3 for this constructive suggestion. When attempting to decrease the sample size, the design of the meandering waveguide is the most crucial factor to be considered. As discussed above, the distance between the coupling points should be greater than the wavelength of the traveling photons ($L > \lambda$) in the ‘giant atom’ configurations. Therefore, we list the following methods for miniaturizing the sample design.

1. Add more bends. We employ fewer bends and a greater distance between the corners compared to the meandering waveguide used in Ref. [8], giving rise to a larger sample size. For the sake of making a smaller sample, we can bend the waveguide more times to miniaturize the sample design.

2. Reduce the effective wavelength of traveling photons in the waveguide. In our waveguide, the wavelength of traveling photons is 2~4 cm at 4~7 GHz. The relative dielectric constant of the waveguide substrate has a significant impact on this value. We can use a substrate with larger

relative dielectric constant to fabricate the waveguide. In this case, the effective wavelength of the photons at a certain frequency is reduced, which can also miniaturize the sample design.

According to referee #3's suggesting, we have added a discussion of how to miniaturize the sample in the section of Conclusion and Outlook on page 5 as follows: "**Moreover, we can miniaturize the sample design by adding more bends along the waveguide or using a substrate with larger relative dielectric constant to fabricate the waveguide.**".

Comment:

•Add a discussion by comparing the results with previous demonstrations (superconducting qubit) that are already cited in the manuscript.

Reply:

We thank referee #3 for this constructive comment. Overall, our findings offer a unique physical platform to realize and demonstrate the mechanism of 'giant atom'. The self-interference effect, which is the defining characteristic of 'giant atom' physics, can be clearly shown in our system. In comparison with the 'giant atom' demonstrations in the superconducting-qubit system (Refs. [6,8,16] in the revised manuscript), the ferromagnetic spin system can be implemented at room temperature and exhibits a large frequency tuning range, simple device fabrication, and the ability to be constructed in a variety of topologies. It is an outstanding new platform for revealing and exploring novel phenomena related to the light-matter interaction beyond the dipole approximation.

Following referee #3's suggestion, we have added the above discussion in the revised manuscript (see the first paragraph on page 6).

Reviewers' Comments:

Reviewer #1:

Remarks to the Author:

I have now read the reply of the authors to both my own referee report and those of referees 2 and 3, and checked the updates to the manuscript. I find that the authors have replied well to all comments from the referees, and made appropriate changes to the manuscript. In light of this, and for the reasons I gave in my previous report, I now believe that the manuscript is ready for publication in Nature Communications. I only ask that the authors consider one last comment from me before publication:

1. In reply to comment 5 in my previous report, the authors have now clarified that the experiments take place at room temperature. However, the relevant transition frequencies in the system are 4-7 GHz. For superconducting qubits to work well at these same frequencies, as for example in the experiments in Refs. [8, 16], the sample needs to be cooled down to millikelvin temperatures to avoid thermal excitations. Why aren't the effects seen in the current experiment washed out by thermal noise? Perhaps this is obvious for someone from the field of magnonics, but for me, who is more used to superconducting qubits, it's puzzling. I think the authors need to comment on this issue in the text to avoid confusion for a large audience.

Reviewer #2:

Remarks to the Author:

The authors responded to my questions in detail, and I appreciate the authors' efforts.

Regarding the significance, I expect that the authors' work can have a wide and solid impact on the magnonics. As the authors stated in main text as well as the response letter, building giant atoms can enhance the traveling-wave-mediated interaction to the strong coupling regime. However, as far as the manuscript shows, the coupling strength between two modes via the mediation of the traveling waves is weak, which limits the usefulness of the "giant spin ensemble" configuration in practical applications. The author believe that their experimental setup is possible to realize the strong coupling, but currently they haven't provided convincing and objective evidence to support their strong statement. I am nervous that it appears to be an overstatement.

In addition, the size of the magnetic spheres is much smaller than the microwave wavelength. Adjusting the design of the waveguide does not change their characteristics as point emitters. The physical process in this work is the superposition of the magnetic dipole interaction and the related interference, which are well-studied knowledge in QED. In my humble opinion, it is not as important as the authors say "beyond the dipole approximation".

Although the authors' introduction of the concept of "giant atoms" into their magnonic device is original and interesting, the current work is still lacking from the breakthroughs of the physical mechanism or the performance of measured coupling effect. Therefore, I foresee a lesser impact in this work and it does not warrant a publication in high level journal as Nature Communications.

Reviewer #3:

Remarks to the Author:

The authors have done a wonderful job in carefully addressing all the comments. The clarity of the manuscript is significantly improved post-revisions. Additional information in the supplementary makes this report further comprehensive. I believe that the results will be of great interest for the magnonics community and beyond. Therefore, I am happy to recommend this work for publication in Nature Communications.

Reply to Referee #1's comments

Comment:

I have now read the reply of the authors to both my own referee report and those of referees 2 and 3, and checked the updates to the manuscript. I find that the authors have replied well to all comments from the referees, and made appropriate changes to the manuscript. In light of this, and for the reasons I gave in my previous report, I now believe that the manuscript is ready for publication in Nature Communications. I only ask that the authors consider one last comment from me before publication:

Reply:

We would like to thank referee #1 again for evaluating our paper and recommending its publication. Below we respond to the additional comment in detail.

Comment:

1. In reply to comment 5 in my previous report, the authors have now clarified that the experiments take place at room temperature. However, the relevant transition frequencies in the system are 4-7 GHz. For superconducting qubits to work well at these same frequencies, as for example in the experiments in Refs. [8, 16], the sample needs to be cooled down to millikelvin temperatures to avoid thermal excitations. Why aren't the effects seen in the current experiment washed out by thermal noise? Perhaps this is obvious for someone from the field of magnonics, but for me, who is more used to superconducting qubits, it's puzzling. I think the authors need to comment on this issue in the text to avoid confusion for a large audience.

Reply:

We thank referee #1 for this insightful comment. The magnon mode is the collective spin excitation mode in the spin ensemble. Using the Holstein–Primakoff transformation, the magnon mode can be treated as a harmonic oscillator mode under the condition of the low-lying excitations. Because the spin density in the yttrium iron garnet (YIG) is very high ($\sim 10^{22} \mu_B \text{ cm}^{-3}$), this condition still allows to have a sufficiently large number of the excitations in the magnon mode. Although a certain number (~ 650) of thermal magnons (10 GHz) are excited at room temperature (300 K), the microwave field in the waveguide can further excite considerable magnons in the Kittel mode because the total number of magnon excitations (including the thermal ones) can be much smaller than the total number of the spins in the YIG sample. Thus, we can ignore the influence of the thermal excitations on the interaction between the magnon mode and microwave photon mode in the waveguide. Indeed, many experiments show that the thermal excitations at room temperature do not give rise to any appreciable backaction or other effects (see, e.g., Ref. [26,27,28,31,36]). In other words, the magnon mode can thoroughly interact with the microwave photons in the waveguide even with certain thermal excitations.

However, the superconducting qubit is a very nonlinear device. Its anharmonicity is sufficiently large and it is therefore be treated as a two-level artificial atom. If the temperature is high, the

qubit is excited by the thermal excitation. Under this circumstance, the traveling photons in the waveguide will not be absorbed by the superconducting qubits any more. Therefore, the further interference effects will be washed out. More importantly, as the referee mentioned, the high temperature will cause a series of noise factors (such as Josephson noise, unpaired electrons, measurement signal noise) to prevent the superconducting qubits from working well.

According to the referee's suggestion, we have added the following description about the magnon mode in the revised manuscript, so as to avoid confusion for a large audience:

“The magnon mode is the collective spin excitation mode, and it can be treated as a harmonic oscillator mode. Since the total number of spins in the YIG sphere is huge [25,26], the Kittel mode can sustain a sufficiently large number of magnon excitations, which is much larger than the thermal excitations at room temperature. Therefore, the thermal excitations will not influence the interaction between the magnon mode and the waveguide photons.”

Reply to Referee #2's comments

Comment:

The authors responded to my questions in detail, and I appreciate the authors' efforts.

Reply:

We would like to thank referee #1 again for evaluating our paper.

Comment:

Regarding the significance, I expect that the authors' work can have a wide and solid impact on the magnonics. As the authors stated in main text as well as the response letter, building giant atoms can enhance the traveling-wave-mediated interaction to the strong coupling regime. However, as far as the manuscript shows, the coupling strength between two modes via the mediation of the traveling waves is weak, which limits the usefulness of the "giant spin ensemble" configuration in practical applications. The author believe that their experimental setup is possible to realize the strong coupling, but currently they haven't provided convincing and objective evidence to support their strong statement. I am nervous that it appears to be an overstatement.

Reply:

We thank referee #2 for his/her comments. The main purpose of our work is to utilize the advantage of waveguide magnonics to experimentally demonstrate a series of striking phenomena of 'giant atom' physics. Our work provides a new and easy-to-operate platform to investigate the light and matter interaction in the context of giant atoms. We have realized periodic coupling and decoupling of the spin ensemble with the waveguide owing to the 'giant atom' physics, frequency-dependent collective behavior of the nested giant spin ensembles, and counterintuitive coherent

interaction between the spin ensembles when one of the giant spin ensembles is decoupled from the waveguide. These advances are considered very innovative and important by both referee #1 and referee #3. In the discussion and outlook section, we have also discussed several potential applications of giant spin ensembles.

For the strong coherent coupling mediated by the meandering waveguide, we have given detailed theoretical analysis in the manuscript. Similar approach has also been experimentally realized in other physical systems [Mirhosseini et al., Nature, 569, 692 (2019)], which confirmed the possibility of realize strong coupling in open waveguides. We also plan to realize the strong coupling in the waveguide magnonics. We believe our work will accelerate the development of ‘giant atom’ physics and endow it with a range of promising applications.

Nevertheless, we have correspondingly toned down the emphasis on strong coupling realization in the present manuscript to avoid possible overstatement.

Comment:

In addition, the size of the magnetic spheres is much smaller than the microwave wavelength. Adjusting the design of the waveguide does not change their characteristics as point emitters. The physical process in this work is the superposition of the magnetic dipole interaction and the related interference, which are well-studied knowledge in QED. In my humble opinion, it is not as important as the authors say "beyond the dipole approximation".

Reply:

We thank referee #2 for his/her comments. The configuration of the two coupling points obviously breaks the dipole approximation effectively, because under the dipole approximation, the atom can only interact with the light field at a single point, which has been clearly explained in the theoretical derivation of our manuscript and described in two other references [Kannan et al., Nature 583, 775-779 (2020); Kockum et al., Phys. Rev. A 90, 013837 (2014)].

Comment:

Although the authors' introduction of the concept of "giant atoms" into their magnonic device is original and interesting, the current work is still lacking from the breakthroughs of the physical mechanism or the performance of measured coupling effect. Therefore, I foresee a lesser impact in this work and it does not warrant a publication in high level journal as Nature Communications.

Reply:

We thank referee #2 for his/her comments. Since the concept of the ‘giant atom’ is proposed several years ago, a series of novel effects arising from the ‘giant atom’ physics have attracted many theorists to investigate this new framework from various aspects (Ref. [17-22]). Due to the system constraints and technical limitations, only superconducting circuits have been utilized to demonstrate the ‘giant atom’ physics until now. The collective behavior between multiple giant atoms was only realized in the braided configuration, as demonstrated in Ref. [8], and the nested giant atoms have never been explored. In the present work, we not only put forward a new and easy-to-operate platform to demonstrate ‘giant atom’ physics, but also realize for the first time

the nested giant spin ensembles. Our main achievements include the realizations of periodic coupling and decoupling of the spin ensemble with the waveguide owing to the ‘giant atom’ physics, frequency-dependent collective behavior of the nested giant spin ensembles, and counterintuitive coherent interaction between the spin ensembles when one of the giant spin ensembles is decoupled from the waveguide. These novelties are highly praised by both referee #1 and referee #3 in their reports.

Through the above explanations, we hope that referee #2 may find our work to be original in terms of exploring the physics of ‘giant atoms’ in waveguide magnonics.

Reply to Referee #3's comments

Comment:

The authors have done a wonderful job in carefully addressing all the comments. The clarity of the manuscript is significantly improved post-revisions. Additional information in the supplementary makes this report further comprehensive. I believe that the results will be of great interest for the magnonics community and beyond. Therefore, I am happy to recommend this work for publication in Nature Communications.

Reply:

We greatly appreciate referee #3 for his efforts in evaluating our revised manuscript and recommending its publication in Nature Communications.